# Enhanced dust removal via the synergy of a standing wave acoustic field and high-pressure spray: An integrated experimental and numerical study

Shixian Wu[1,2,3], Hui Zhu[1]*, Can Qi[4], Yongping Chen[4], Hui Yang[4], Chunyu Liu[4], Shiqiang Chen[3], Heqing Liu[3]

1 School of Architecture and Transportation Engineering, Guilin University of Electronic Technology, Guilin, China, 2 School of Green Building and Low-Carbon Technology, Guangxi Technological College of Machinery and Electricity, Nanning, China, 3 School of Resources, Environment and Safety Engineering, Hunan University of Science and Technology, Xiangtan, China, 4 Guangxi Key Laboratory of Clean Energy Equipment and Energy Saving Technology for Higher Education Institutions, Guilin University of Aerospace Technology, Guilin, China

* zhuhui@guet.edu.cn

## Abstract

Occupational exposure to respirable coal dust poses severe health risks in underground mining operations, primarily through the development of coal workers' pneumoconiosis (CWP)—a progressive and irreversible pulmonary disease. To address this challenge, we developed an innovative dust suppression system that integrates ultrasonic atomization with acoustic agglomeration technology. The system operates via a dual-phase mechanism: ultrasonic atomization generates ultrafine water droplets (<10 μm) to form a heterogeneous dust-droplet dispersion, while high-frequency standing wave fields (20 kHz) concentrate airborne particles spatially, thereby enhancing interphase collisions between droplets and dust. The resulting agglomerates are subsequently removed by a high-pressure spray. System performance was systematically evaluated through scaled laboratory experiments that examined three critical operational parameters: nozzle orifice diameter (0.4–0.8 mm) of the high-pressure spray system, acoustic power density (60–180 W) utilized to generate the standing wave field, and duct airflow velocity (0.25–0.75 m/s). Numerical simulations integrating acoustic dynamics with CFD-DEM modeling were employed to elucidate particle trajectories and the spatiotemporal evolution of dust-droplet agglomerates within the coupled acoustic-flow field. Experimental results demonstrate a greater than 10% improvement in $PM_{2.5}$ removal efficiency compared to conventional high-pressure spray systems. This enhancement is attributed to the synergistic effects of acoustic focusing and droplet entrapment. The study establishes a foundational framework for the development of acoustically enhanced pretreatment systems and offers a practical strategy for reducing respirable dust exposure in underground mining environments.

**Data availability statement:** All relevant data are within the paper and its Supporting Information file.

**Funding:** This work was supported by the National Natural Science Foundation of China (NSFC), Grant 52264017 (to HZ and SW), and the Natural Science Foundation of Guangxi Zhuang Autonomous Region, Grant 2021GXNSFAA220079 (to HZ and SW). The funders had no role in the study design, data collection and analysis, decision to publish, or manuscript preparation.

**Competing interests:** The authors have declared that no competing interests exist.

## 1. Introduction

Coal, as a non-renewable energy resource, continues to dominate global primary energy consumption, accounting for over 30% of the total energy mix [1–3]. The widespread adoption of mechanized coal mining technologies has substantially increased coal dust concentrations in underground operations, posing severe health risks through prolonged occupational exposure [4–6]. Furthermore, dust generated during coal fragmentation exhibits enhanced oxygen accessibility, making suspended coal particles highly explosive under elevated thermal conditions [7–9]. These dual threats establish coal dust control as a critical priority in mining safety management, with direct implications for both operational continuity and workforce well-being.

Contemporary dust mitigation strategies in major coal-producing nations employ three primary approaches. The first involves enhancing the wettability of the coal mass by water injection prior to coal mining, thereby reducing dust generation potential [10]. The second approach focuses on controlling dust at the source within ventilated spaces using specific dust-suppression technologies, such as spray dust reduction and foam dust removal [11,12]. The third strategy utilizes dust collectors to filter or eliminate dust promptly. These collectors can be categorized into dry and wet types. Common dry dust collectors used underground include inertial and cyclone collectors, which typically achieve total dust removal efficiencies of 80–95%. A cyclone dust collector separates dust particles from airflow via centrifugal force, trapping them on the device walls [13,14], while an inertial dust collector uses particle inertia to capture dust through impaction on baffles [15,16]. Dry filtration dust collectors employ fibrous media to filter dust from the air [17−18]. Wet dust collectors utilize water or other liquids to interact with dusty air through droplets, films, or bubbles, facilitating dust capture via collision and coagulation [19–22]. These systems offer low explosion risk and can handle high-temperature, flammable, and explosive gases, with total dust removal efficiency generally exceeding 90%. Their effective secondary dust prevention and compact design further enhance their practicality [23].

Despite these advancements, current dust prevention and control technologies face several challenges. Coal-seam water injection is less effective in low-porosity or resistant coal seams. Spray droplets in complex mining environments show significant variability, leading to unstable atomization performance and reduced respirable dust removal efficiency. Dry inertial and cyclone collectors also exhibit limited effectiveness in capturing respirable dust. Moreover, dry filter dust collectors are unsuitable for high-humidity conditions. Conventional spray systems used in dust collectors demonstrate suboptimal performance in controlling respirable dust ($PM_{10}$), with field efficiencies ranging only from 40% to 60% [24].

Current research on dust suppression focuses on two optimization strategies: (1) enhancing atomization through nozzle redesign and pressure modulation to produce finer droplets (<50 µm) [25–27], and (2) modifying interfacial properties using surfactant additives and magnetized water to strengthen dust-droplet interactions [28–30]. While these approaches demonstrate moderate efficacy for coarse particles (>10 µm), their performance declines significantly for respirable dust ($PM_{10}$), with field

studies reporting capture efficiency ≤ 45%. Therefore, respirable dust control remains a significant technical challenge, necessitating exploration of alternative approaches.

Acoustic agglomeration has emerged as a novel particle manipulation technique, with successful applications in coal-fired flue gas purification, smoke suppression, and precipitation enhancement [31–35]. This technology utilizes acoustic-induced particle migration and collision mechanisms to form rapidly settling aggregates. Standing wave configurations are particularly advantageous due to stable nodal accumulation and lower power consumption. Mining environments offer inherent benefits for implementation: tunnel geometries naturally form standing wave chambers, high dust concentrations increase collision probabilities, and spray-induced humidity levels above 80% promote liquid-bridging agglomeration [36–38]. Although previous studies have explored acoustic agglomeration enhanced by "seed particles," limited data exist on the dust removal efficiency of standing wave acoustic fields combined with spraying. Moreover, the agglomeration mechanisms of ultrafine droplets and particles under coupled acoustic-flow fields remain unclear.

In this study, we experimentally evaluated two synergistic dust removal strategies: (1) high-frequency acoustic agglomeration of coal dust as a pretreatment before high-pressure spray, and (2) ultrasonic atomization to generate ultrafine droplets as "agglomeration nuclei," followed by "coal dust-droplet" acoustic agglomeration and high-pressure spray. Numerical simulations of particle-droplet dynamics in acoustic-flow fields were conducted to elucidate the experimental results. This work advances fundamental understanding of acoustic-spray synergy and assesses its potential for engineering applications.

## 2. Materials and methods

### 2.1. Experimental study

**2.1.1. Experimental design.** Multiple potential synergistic strategies exist between sound waves and sprays. Based on preliminary trials of numerous schemes, two effective synergistic approaches were selected for further experimental investigation and analysis.

**Scheme** I: High-frequency acoustic agglomeration of coal dust was used as a pretreatment step prior to dust removal via high-pressure spray, as illustrated in Fig 1(a).

**Scheme** II: An ultrasonic atomizer was employed to generate ultrafine droplets upstream of the acoustic agglomeration chamber. These droplets functioned as "agglomeration nuclei," introduced into the chamber to enhance the acoustic agglomeration of coal dust with droplets before high-pressure spray removal, as shown in Fig 1(b).

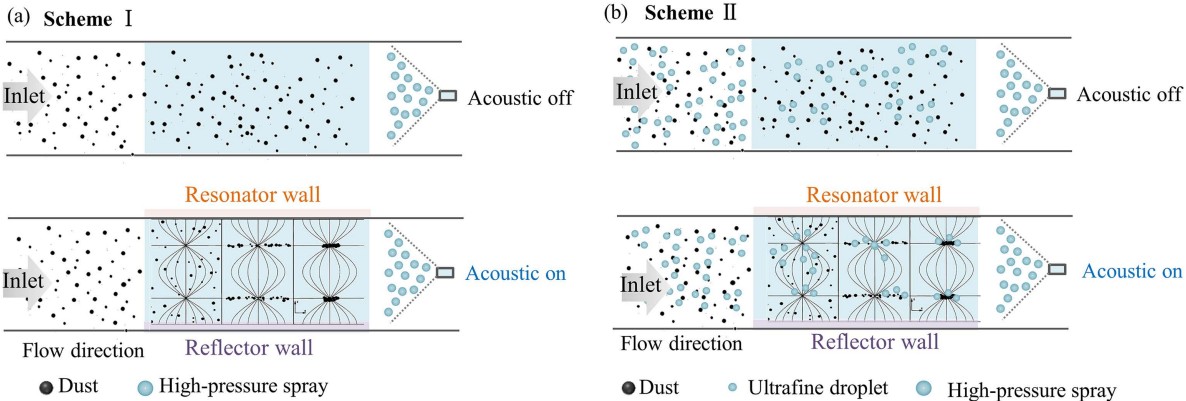

**Fig 1. Two synergistic schemes for dust removal. (a)** Scheme I: acoustic pretreatment + high-pressure spray; **(b)** Scheme II: ultrafine droplet nucleation + acoustic agglomeration + high-pressure spray.

**2.1.2. Experimental setup.** The experimental setup, illustrated in Fig 2, consists of five main components: 1) a scaled tunnel prototype, 2) dust dispersion equipment, 3) an acoustic generation system, 4) an ultrasonic atomization device, and 5) measurement instrumentations. The rectangular tunnel (3.5 m length × 0.2 m × 0.2 m cross-section) incorporates four functional segments sequentially along the airflow path: an ultrasonic nebulization section, an acoustic agglomeration chamber, a pressurized spray section, and a ventilation unit. Transparent acrylic panels were used in tunnel construction to allow optical access for flow visualization and droplet characterization. The dust removal mechanism, which combines acoustic pretreatment and spray capture, operates as follows: particulate matter undergoes enhanced coagulation in the resonant chamber through orthokinetic and hydrodynamic interactions prior to final removal via high-pressure spraying. The explosion-proof dust samplers, FCC-25, were used to sample the dust in the two measuring sections in the roadway model, i.e., one before spray and one after the spray. Under each working condition, three continuous measurements were used to obtain the average value. The dust was weighed with filter membrane by an electronic analytical balance before and after sampling to calculate the total dust removal efficiency.

**2.1.3. Coal dust characteristics.** The experimental coal dust was prepared by crushing anthracite coal blocks and sieving the particles through a 500-mesh (31 μm) industrial sieve. The size distribution of respirable dust was quantified using the LS13320 laser particle analyzer. As shown in Fig 3, the particle size distribution analysis indicated that 68% of the dust particles are below 10 μm, with the maximum particle size not exceeding 33 μm. This size distribution meets the requirements for studying respirable coal dust removal efficiency.

**2.1.4. Droplet atomization characteristics.** In this study, the droplets involved include ultrafine droplets serving as agglomeration nuclei for acoustic agglomeration of particles, as well as droplets generated by the high-pressure spray system. The ultrafine droplets were produced by an ultrasonic atomizer (DOROSIN, China). The size distributions of both types of droplets were analyzed with a Laser Diffraction System (Insitec®, Malvern Instruments, UK). The ultrafine droplets exhibited a size range of 0.1 μm to 14.5 μm, which is comparable to the size of the coal dust particles used in the experiments.

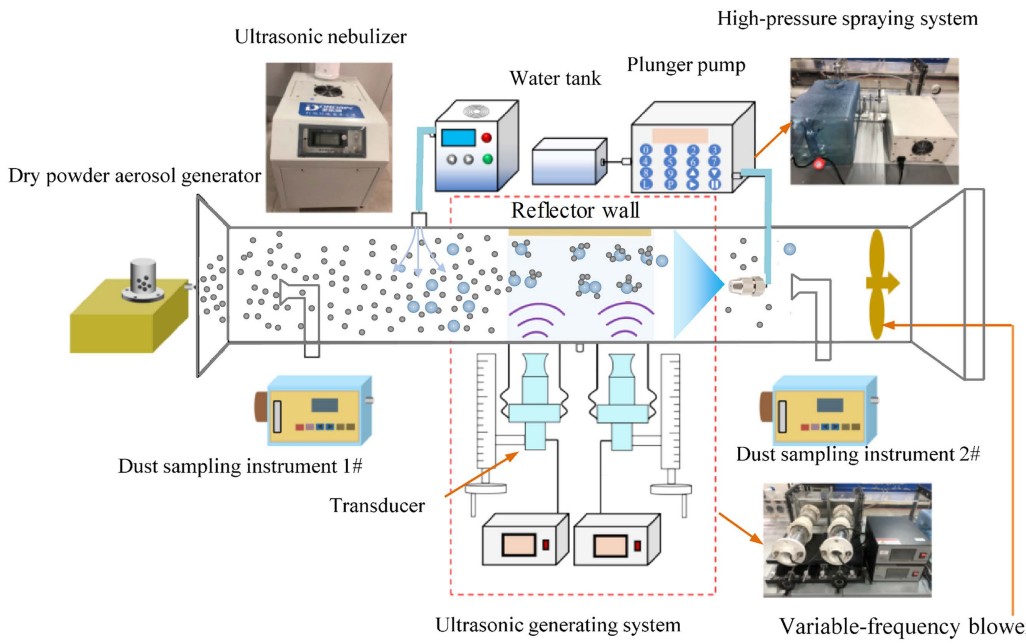

**Fig 2. Schematic diagram of the experimental setup for acoustic-spray synergistic dust removal.**

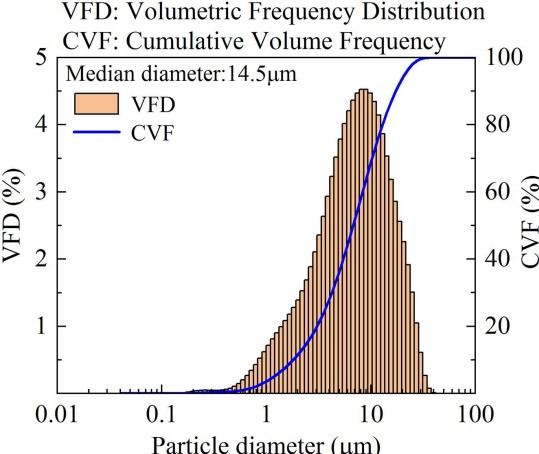

**Fig 3. Particle size distribution of the experimental coal dust.**

For the high-pressure spray, the droplet size distribution and spray angle are key parameters influencing droplet–dust interactions. In this investigation, three nozzles with diameters of 0.4 mm, 0.6 mm, and 0.8 mm were selected. The water flow rate was maintained at 180 mL/min, at which the generated spray provided adequate coverage of the experimental roadway model. In practical spray applications, the spray angle determines the effective dust capture area of the spray field—a wider angle corresponds to a larger volume of dust-laden air that can be treated. Fig 4 shows the spray angles for the three nozzle orifice sizes (0.4 mm, 0.6 mm, and 0.8 mm). As observed, the spray angle formed by the 0.4 mm nozzle is approximately 70°, while the 0.8 mm nozzle produces a spray angle of about 90°, indicating a larger effective dust capture area.

The droplet size distribution for the different nozzle orifice sizes is shown in Fig 5. As the nozzle diameter decreases, the median droplet diameter shifts towards smaller droplets. Specifically, the median droplet diameter is approximately 50 μm for the 0.4 mm nozzle, compared to about 90 μm for the 0.8 mm nozzle. In this study, the droplet

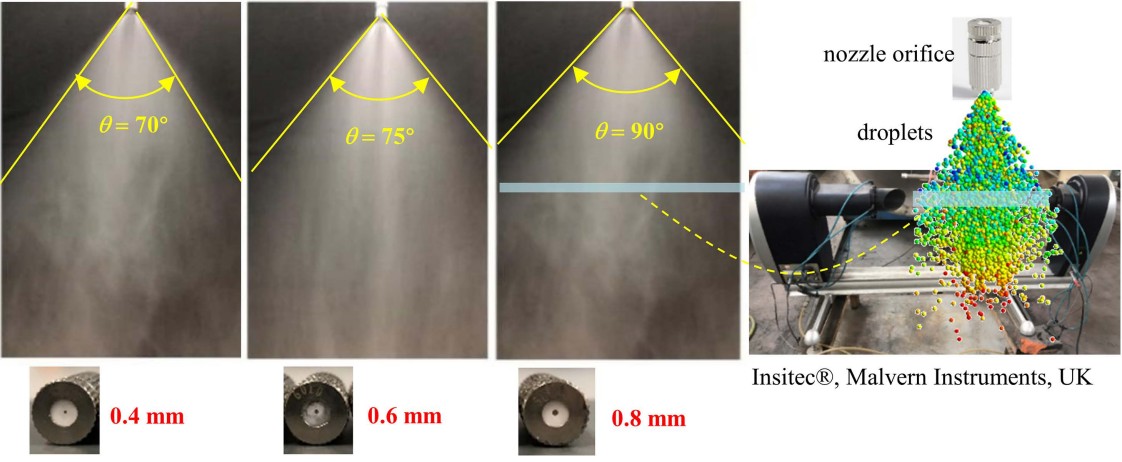

**Fig 4. Spray angle for three nozzle orifice sizes.**

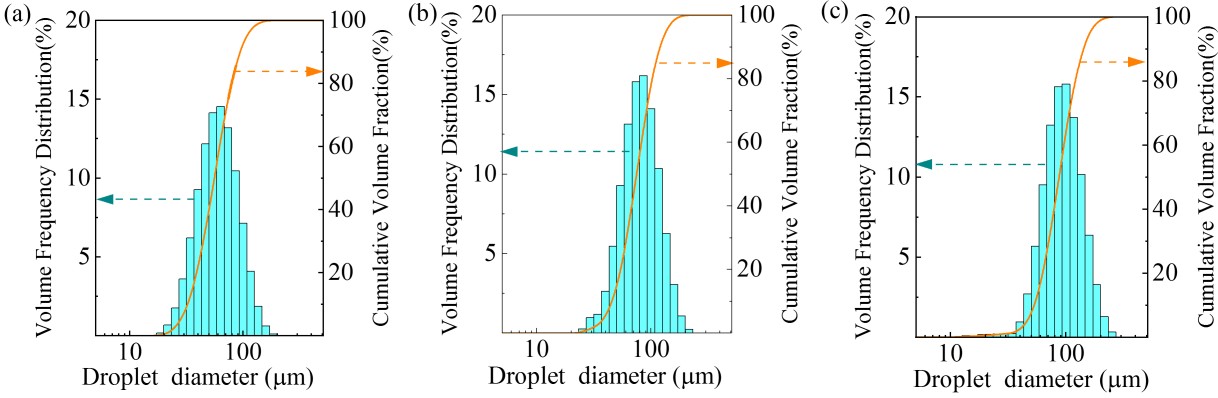

**Fig 5. Droplet size distribution in high-pressure water spray.**

size distribution produced by these nozzles is similar to that generated by high-flow nozzles used in practical engineering applications [39,40].

**2.1.5. Experimental protocol.** The Experimental Scheme I employed a two-stage comparative methodology to evaluate the acoustic-enhanced dust removal mechanism, as illustrated in Fig 1(a). The investigation comprised sequential operational modes: 1) Baseline configuration (Dataset 1): With the acoustic generator deactivated, dust removal efficiency attributable solely to the high-pressure spraying system was recorded during coal dust passed through the roadway model, 2) Acoustic-coupled configuration (Dataset 2): Under identical hydrodynamic conditions, the acoustic generator was activated at a frequency of 20 kHz, enabling particle acoustic agglomeration prior to spray capture. This operation represents the combined efficiency of acoustic preconditioning and subsequent dust removal by spraying. Detailed experimental parameters for Scheme I are summarized in Table 1.

The second synergistic approach (Scheme II, see Fig 1(b)) incorporates an ultrasonic atomization stage upstream of the agglomeration chamber to produce ultrafine droplets (10–50 μm) functioning as "agglomeration nuclei," followed by a combined process of dust-ultrafine droplet acoustic agglomeration and high-pressure spray. This scheme was evaluated using a two-phase protocol consisting of baseline measurements (Dataset 3) to record dust removal from droplet-spray interaction alone, and treatment measurements (Dataset 4) to assess system performance under activated acoustic-droplet-dust agglomeration. while cross-comparison with Dataset 2 from Scheme I helped identify the optimal acoustic-spray configuration for industrial dust control applications.

Existing studies indicate that low-frequency sound waves are more effective in promoting particle agglomeration [31]. However, considering the potential health impacts on personnel, this study employed high-frequency ultrasound (minimum 20 kHz), which is inaudible to the human ear. The ultrasonic generator used was a commercially available standard device (not custom-developed) with a safe operating power range of 60–180 W. It should be noted that the actual factor

**Table 1. Experimental parameters for experimental Scheme I.**

| Controlled Variables | Tested Variables | Parameter Values |
|---|---|---|
| $f = 20$ kHz, $P = 180$ W, $Q_w = 180$ mL/min | Air velocity | 0.25 m/s |
| $f = 20$ kHz, $u_0 = 0.25$ m/s, $Q_w = 180$ mL/min | Acoustic power | 180W, 120W, 60W |
| $f = 20$ kHz, $u_0 = 0.25$ m/s, $P = 180$ W | Spray flow rate | 180 mL/min |

*Notation: $f$ (Acoustic frequency), $P$ (Acoustic power), $u$ (Air velocity), $Q_w$ (Spray flow rate).

influencing agglomeration is the acoustic energy intensity, which is indirectly reflected by the generator power. The acoustic power required in practice depends on the size of the treatment space and the acoustic characteristics of the boundaries. The detailed experimental parameters provided in Table 2.

## 2.2. Numerical modelling

### 2.2.1. Model description.
We developed a multiphase computational fluid dynamics (CFD) model to investigate particle-droplet interaction dynamics within a standing wave acoustic field. As shown in Fig 6, the simulation domain comprises a constrained computational geometry measuring 700 mm ($L$) × 31 mm ($W$) × 2$\lambda$ ($H$), where $\lambda$ represents the acoustic wavelength. The 500 mm active acoustic zone incorporates two critical boundary configurations: (1) a sound-hard reflector at the bottom surface, and (2) a tunable wave source (20 kHz frequency) at the top boundary. The computational approach implemented three critical modeling assumptions: 1) axial wave propagation along the duct's $y$-axis generating stationary wave patterns, 2) negligible acoustic energy attenuation in the resonant chamber, and 3) isothermal flow conditions. This controlled simplification strategy preserved the physical mechanisms governing aerosol transport while

**Table 2. Experimental parameters for experimental Scheme II.**

| Controlled Variables | Tested Variable | Parameter Values |
|---|---|---|
| $f$ = 20 kHz, $u_0$ = 0.25 m/s, $P$ = 180 W, $Q_w$ = 180 mL/min | Atomizer flow rate | 10 mL/min |
| $f$ = 20 kHz, $u_0$ = 0.25 m/s, $Q_w$ = 180 mL/min, $Q_{atom}$ = 10 mL/min | Acoustic power | 180 W, 120 W, 60 W |
| $f$ = 20 kHz, $u_0$ = 0.25 m/s, $P$ = 180 W, $Q_{atom}$ = 10 mL/min | Spray flow rate | 180 mL/min |
| $f$ = 20 kHz, $Q_w$ = 180 mL/min, $Q_{atom}$ = 10 mL/min | Air velocity | 0.25 m/s |

\*Notation: $f$ (Acoustic frequency), $P$ (Acoustic power), $u$ (Air velocity), $Q_w$ (Spray flow rate), $Q_{atom}$ (Atomizer flow rate).

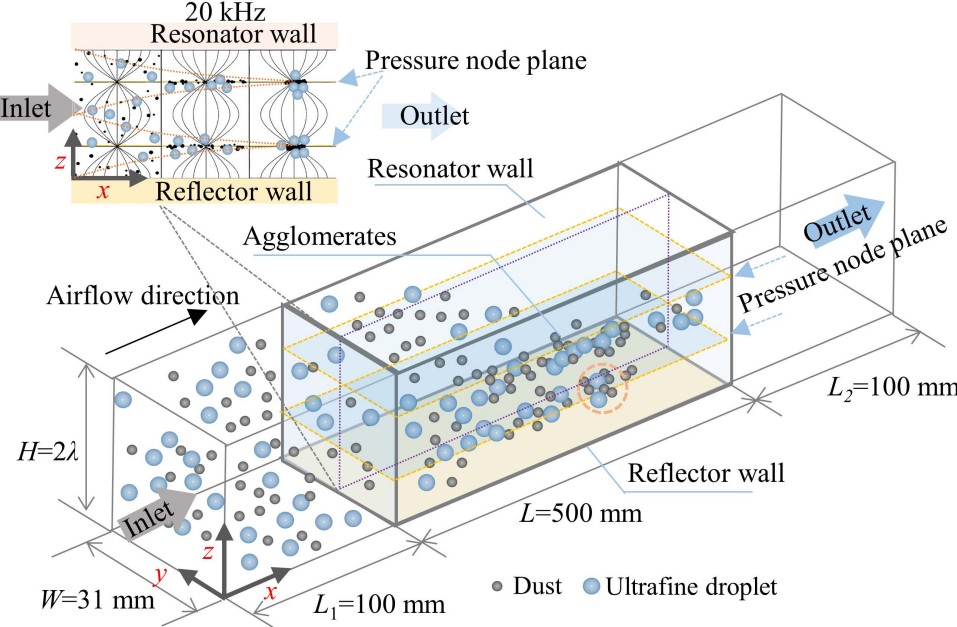

**Fig 6. Numerical simulation model for particle-droplet interaction dynamics in a standing-wave acoustic field.**

ensuring experimental parameter alignment for investigating fundamental particle-droplet interactions under combined acoustic-flow field.

**2.2.2. Dynamics of dust/droplet motion.** The standing-wave acoustic field can be described by the following wave equation [41]

$$u_f(y, t) = u_a \sin(ky) \cos(\omega t) \tag{1}$$

where, $u_a$ is the velocity amplitude, $k$ is the wavenumber, defined as $k = \omega/c = 2\pi/\lambda$, $c$ is the speed of sound, $\omega$ is the angular frequency, $f$ is the acoustic wave frequency, $t$ is time, $y$ is the spatial coordinate.

Given that the turbulent fluctuations in the roadway model are far more intense than particle Brownian motion, the latter's effect was neglected. Consequently, the motion of dust particles and droplets is governed by the following equation, which accounts for viscous entrainment and unsteady forces:

$$m_p \frac{du_p}{dt} = F_{st} + F_B + F_p + F_{vm} + \langle F_{ac} \rangle \tag{2}$$

where, $m_p$ is particle mass, $u_p$ is particle velocity, $F_{st}$ is Stokes drag force, $F_B$ is Basset history force, $F_p$ is pressure gradient force, $F_{vm}$ is virtual mass force, $F_{ac}$ is time-averaged acoustic radiation force.

The individual force components are given by [42,43]

$$F_{st} = 3\pi \mu_g d_p (u_f - v_p)/C_m \tag{3}$$

$$F_B = \frac{3}{2} d_p^2 \sqrt{\pi \rho_g \mu_g} \int_0^t \left( \frac{du_f}{dt'} - \frac{dv_p}{dt'} \right) \frac{1}{\sqrt{t - t'}} dt' \tag{4}$$

$$F_p = m_f \left( \frac{\partial u_f}{\partial t} + u_f \frac{\partial u_f}{\partial x} \right) \tag{5}$$

$$F_{vm} = \frac{1}{2} m_f \left( \frac{du_f}{dt} - \frac{dv_p}{dt} \right) \tag{6}$$

where, $\mu_g$ is fluid dynamic viscosity, $d_p$ is particle diameter, $v_p$ and $u_f$ are particle and fluid velocity, respectively, $\rho_g$ is fluid density, $t$ is time variable, $m_f$ is the mass of the fluid that has the same volume as the particle.

For the planar standing wave acoustic field shown in Fig 6, the acoustic radiation force ($F_{ac}$) acting on a particle is calculated as [44]:

$$\langle F_{ac} \rangle = F_{max} \sin(2k \cdot y) \cdot j \tag{7}$$

where, $\langle \cdot \rangle$ is time-averaging operator over acoustic periods, $F_{max}$ is maximum acoustic radiation force, $y$ is spatial coordinate along wave propagation direction, $j$ is unit vector in $y$-direction (i.e., wave propagation direction).

$$F_{max} = \frac{1}{2} \pi k E_{ac} d_p^3 \cdot \Phi \tag{8}$$

$$\Phi = \frac{1}{3} f_1 \left( \frac{\kappa_p}{\kappa} \right) + \frac{1}{2} f_2 \left( \frac{\rho_p}{\rho_g}, \frac{2\delta}{d_p} \right) \tag{9}$$

$$f_2\left(\frac{\rho_p}{\rho_g}, \frac{2\delta}{d_p}\right) = \text{Re}\left[2\left[1 - \gamma\left(\frac{2\delta}{d_p}\right)\right]\left(\frac{\rho_p}{\rho_g} - 1\right)\Big/\left[2\frac{\rho_p}{\rho_g} + 1 - 3\gamma\left(\frac{2\delta}{d_p}\right)\right]\right] \tag{10}$$

$$f_1\left(\frac{\kappa_p}{\kappa}\right) = 1 - \frac{\kappa_p}{\kappa}; \quad \gamma\left(\frac{2\delta}{d_p}\right) = -\frac{3}{2}\left[1 + i\left(1 + \frac{2\delta}{d_p}\right)\right]\frac{2\delta}{d_p} \tag{11}$$

where, $E_{ac}$ is acoustic energy density, $\Phi$ is acoustic contrast factor, $\kappa_p$ and $\kappa$ are particle and fluid compressibility, respectively, Re is real part operator.

The acoustic boundary layer thickness ($\delta$) is defined as:

$$\delta = \sqrt{2\nu/\omega} \tag{12}$$

where $\nu$ is kinematic viscosity of fluid. The particle motion equation (Eq. 2) is solved using an adaptive step-size fourth-order Runge-Kutta method.

**2.2.3. CFD-DEM modeling of dust-droplet agglomeration.** Tracking individual particles in acoustic-flow fields is computationally prohibitive for systems exceeding $10^7$ particles. To overcome this limitation, a Representative Particle Cluster (RPC) mode that analogous to fluid parcels was developed in ANSYS. In this approach, each computational particle represents an agglomerate of physical particles, with mass and momentum conservation maintained. This method enhances computational efficiency by a factor of $10^2$ to $10^3$ while preserving the key physical phenomena such as acoustic focusing, size-dependent agglomeration, and flow transport. The collision dynamics between dust-droplet clusters were modeled using a momentum-based approach derived from Newton's second law. The coalescence or rebound of two particle clusters upon collision between two particle clusters is determined by Eq. 13.

$$\left|\vec{v}_{p1} - \vec{v}_{p2}\right| < v_{cr} \tag{13}$$

where: $v_p$ is the normal relative velocity between particle clusters (m/s), $v_{cr}$ is the critical coalescence velocity, calculated from energy balance principles [43−45].

When Eq. (13) is satisfied, the particle clusters coalesced with a resultant velocity given by

$$v_{agg} = \frac{m_{p1}v_{p1} + m_{p2}v_{p2}}{m_{p1} + m_{p2}} \tag{14}$$

where, $v_{agg}$ is the velocity of the agglomerate, $m_{p1}$ and $m_{p2}$ are the masses of particle clusters P1 and P2, $v_{p1}$ and $v_{p1}$ are the pre-collision velocities of P1 and P2. In the numerical simulations, both dust particles and droplet particles were initialized with a uniform spatial distribution at the inlet of the agglomeration chamber. Their velocity magnitudes were set equal to the local airflow velocity at their respective positions. A frozen-wall boundary condition was implemented, treating particle-wall collisions as irreversible deposition events and terminating further trajectory calculations upon impact. This method assumed complete suppression of rebound mechanisms and disregarded particle resuspension under the simulated conditions.

The flow field was solved using ANSYS Fluent within the computational domain illustrated in Fig 4. To ensure accuracy, a structured mesh with local refinement near the walls was employed. The particle dynamics model was implemented through User-Defined Functions (UDFs). All simulation parameters are summarized in Table 3.

Due to negligible acoustic energy dissipation in the model (Fig 4), all particles exhibited consistent migration behavior toward pressure nodes. This uniformity allowed the analysis to focus on a representative pressure node region, where the influence of key parameters on particle trajectories were investigated.

**Table 3. Parameter values for numerical calculation.**

| Parameter | values |
|---|---|
| Ultrasonic frequency $f$ | 20–50 kHz |
| Sound propagation speed $c$ | 340 m/s |
| Acoustic energy density $E_{ac}$ | 0.5–10 J/m³ |
| Particle diameters $d_p$ | 1–10 μm |
| Ultrafine droplet diameters $d_w$ | 10–50 μm |
| Air density $\rho_g$ | 1.2 kg/m³ |
| Particle densities $\rho_p$ | 2000 kg/m³ |
| Droplet densities $\rho_w$ | 1000 kg/m³ |
| Air compressibility coefficients $\kappa$ | $7.143 \times 10^{-6}$ Pa⁻¹ |
| Particle compressibility coefficients $\kappa_p$ | $2.78 \times 10^{-11}$ Pa⁻¹ |
| Droplet compressibility coefficients $\kappa_p$ | $5 \times 10^{-10}$ Pa⁻¹ |
| Air kinematic $\nu$ | $1.51 \times 10^{-5}$ m²/s |
| Dynamic viscosities $\mu_g$ | $1.82 \times 10^{-5}$ Pa·s |

## 3. Results and discussion

### 3.1. Experimental study

**3.1.1. Effects of nozzle orifice diameter.** The influence of nozzle orifice diameter on total dust removal efficiency is given in Fig 7(a) for **Scheme I** (acoustic waves + high-pressure spray). The experimental parameters were set as follows: acoustic power $P = 180$ W, duct airflow velocity $u_g = 0.25$ m/s, ultrasonic atomizer flow rate $Q_{atom} = 10$ mL/min and high-pressure spray flow rate $Q_w = 180$ mL/min. The combination of acoustic waves and high-pressure spray yielded only a marginal improvement in overall dust removal efficiency. This suggests that the synergistic effect in **Scheme I** is limited under the tested conditions.

Under identical operating parameters, **Scheme II**, which synergistically combines ultrafine droplets, acoustic waves, and a high-pressure spray, yielded a marked performance improvement, as evidenced by the data in Fig 7(b). A comparative analysis of the performance enhancement relative to a standalone high-pressure spray system is summarized in Fig 7(c). **Scheme II** consistently outperformed **Scheme I** across nozzle orifice diameters ranging from 0.4 mm to 0.8 mm, with a maximum efficiency improvement of 11% observed at 0.8 mm. This represents a 5.9-fold increase over the meager

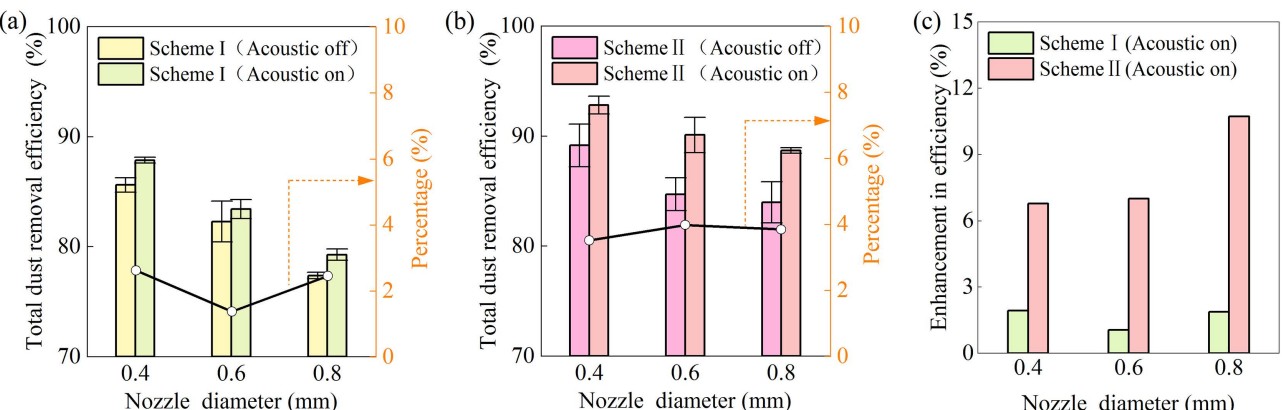

**Fig 7. Dust removal efficiency as a function of nozzle orifice diameter.**

1.86% enhancement achieved by **Scheme I**. The marked superiority of Scheme II underscores the essential function of ultrafine droplets ($d_w < 10$ μm) serving as "agglomeration nuclei" within the acoustic chamber. These results affirm a strong synergy among ultrasonic atomization, acoustic agglomeration, and high-pressure spray—a finding with promising implications for controlling respirable dust in mining environments.

**3.1.2. Effects of acoustic power.** The role of acoustic power in enhancing dust removal across different particle size ranges is illustrated in Fig 8. Experiments were performed with a duct airflow velocity $u_g = 0.25$ m/s, ultrasonic atomizer flow rate $Q_{atom} = 10$ mL/min, and high-pressure spray flow rate $Q_w = 180$ mL/min. The horizontal axis denotes the upper limit of each particle size bin (e.g., 5 μm corresponds to the 0–5 μm).

As shown in Fig 8(a), effective acoustic agglomeration was only observed beyond a threshold acoustic power level, irrespective of particle size. This phenomenon is attributed to the strong turbulent flow in the agglomeration zone, where turbulent diffusion disrupts particle motion and counteracts agglomeration. Below the critical power level, the acoustic radiation force is insufficient to overcome this turbulent diffusion, preventing the particle concentration from reaching the level necessary for successful agglomeration. Consequently, the potential benefit of acoustic waves in improving spray dust removal remains unrealized under such conditions.

Fig 8(b) quantifies the enhancement in dust removal efficiency with increasing acoustic power. At 180 W, removal efficiency for PM$_{2.5}$ improved by approximately 10%. However, operating at such power intensities also leads to substantial thermal loading on the transducers, implying higher energy consumption—a relevant factor for practical system design.

**3.1.3. Effects of airflow velocity.** The airflow velocity directly determines the effective duration of the acoustic field acting on the particles (dust or droplets), thereby influencing the actual performance of the acoustic-spray synergistic dust removal. Fig 9 presents the measured total dust removal efficiency under three different airflow velocities with the combined acoustic and spray system. The experimental parameters were set as follows: spray water flow rate of 180 mL/min, ultrasonic atomizer water flow rate of 10 mL/min, and acoustic power of 180 W.

As shown in the Fig 9, an increase in roadway air velocity leads to a noticeable decrease in total dust removal efficiency, regardless of whether acoustic waves are applied, accompanied by a progressive weakening of the synergy between acoustic agglomeration and spray capture. At 0.25 m/s, a significant synergistic effect is observed, enhancing the dust removal efficiency by approximately 7% compared to the scenario without acoustic excitation. However, as the velocity increases to 0.75 m/s, the synergistic improvement virtually disappears.

To further examine the dust suppression performance of the acoustic-spray synergy on dust particles of different sizes, Fig 10 presents the relationship between dust removal efficiency and particle size under different airflow

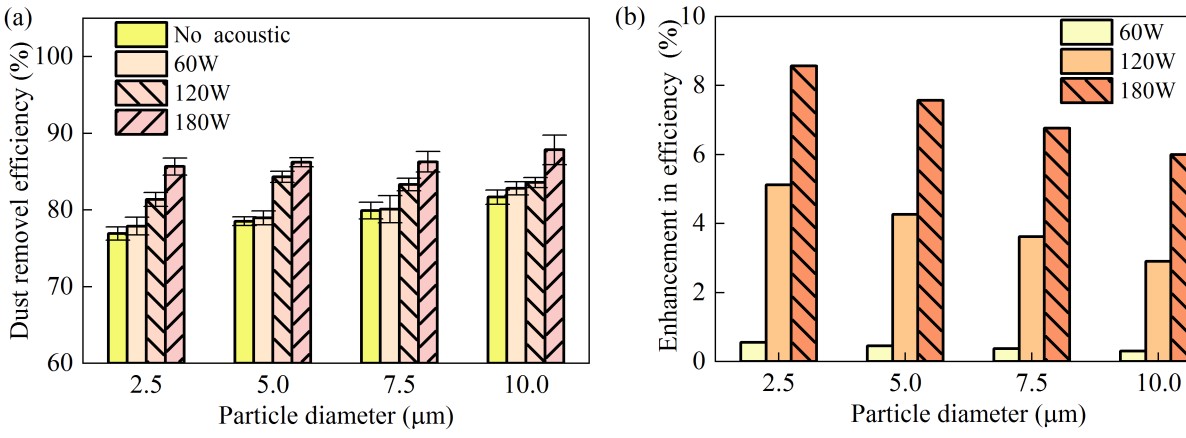

**Fig 8. Dust removal efficiency as a function of acoustic power.**

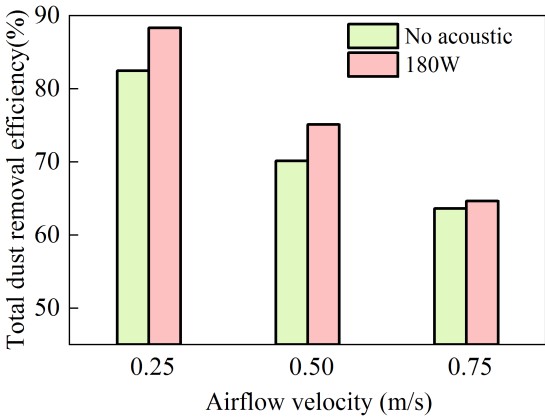

**Fig 9. Dust removal efficiency as a function of air velocity.**

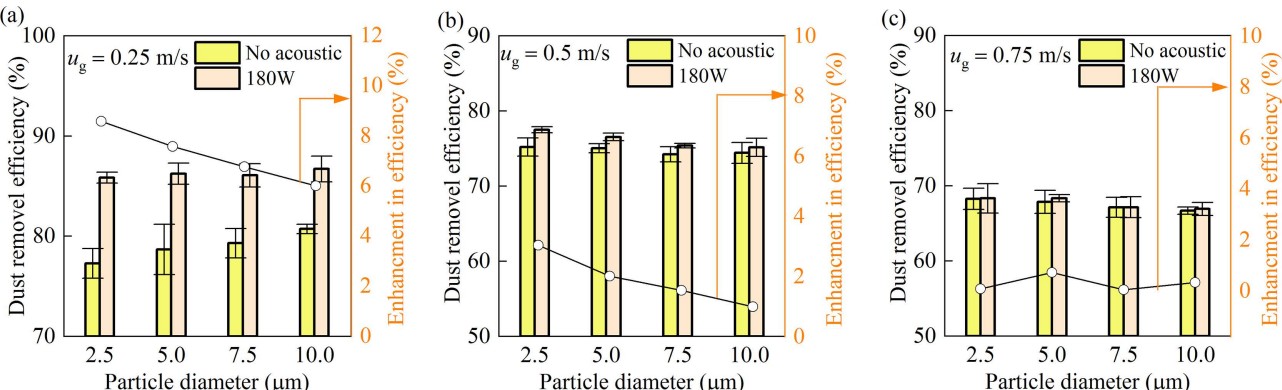

**Fig 10. Removal efficiency as a function of coal dust particle size at different air velocities.**

velocities. At the lowest velocity of 0.25 m/s (Fig 10 (a)), the synergistic effect enhances the capture of particles across the entire size spectrum, with a particularly notable improvement for finer particles: the removal efficiency increased by 6.25% for $PM_{10}$ and 8.57% for $PM_{2.5}$. In contrast, at the higher velocity of 0.75 m/s (Fig 10 (c)), the efficiency improvement for all particle sizes falls below 1%, indicating that the practical synergistic effect is largely lost under high-flow conditions.

Particle removal efficiency depends on whether the particle residence time within the acoustic zone is long enough to overcome turbulent dispersion. This explains the preferential $PM_{2.5}$ removal at low flow rates, where extended interaction time enables effective acoustic agglomeration. Consequently, a key operational guideline for maximizing synergy is to reduce airflow velocity, thereby prolonging the time particles remain in the treatment zone.

### 3.2. Numerical analysis

**3.2.1. Particle motion trajectories.** Fig 11 presents the trajectory spectra of particles with two typical diameters ($d_p$ = 2.5 μm and 10 μm) under varying acoustic energy densities. The simulation parameters were as follows: duct flow velocity $u_g$ = 0.75 m/s, ultrasonic frequency $f$ = 20 kHz, and the acoustic wave propagates along the $y$-direction to form a standing wave field. The trajectories were depicted within a half-wavelength region ($\lambda/2$ = 8.5 mm, where $\lambda$ denotes the

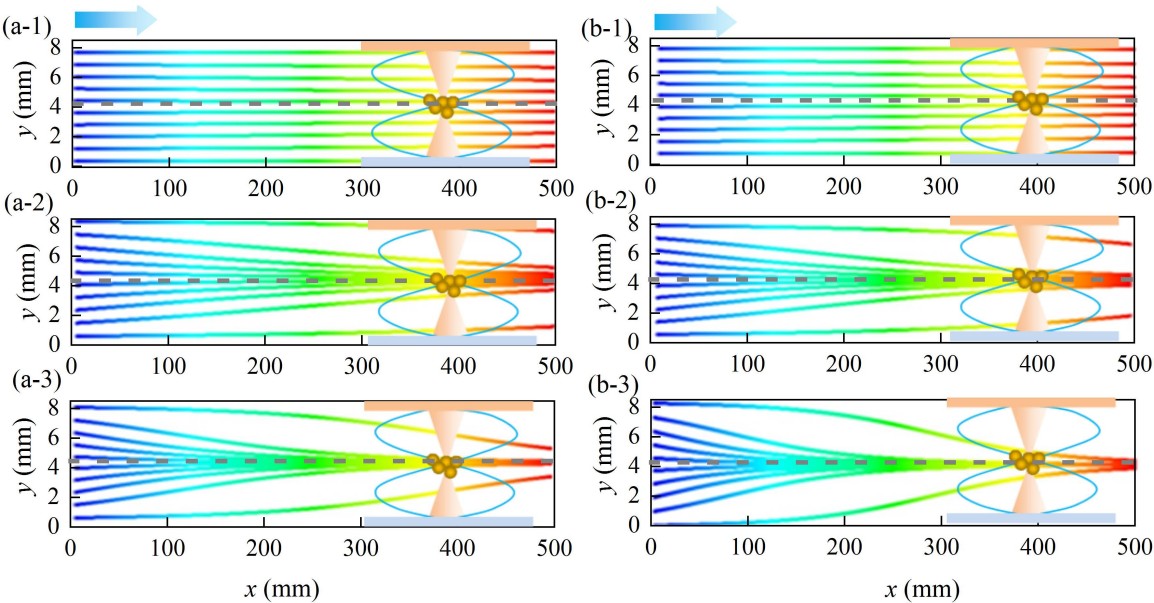

**Fig 11. Trajectories of dust particles under different ultrasonic energy densities.** (a-1) $d_p = 2.5$ µm, $E_{ac} = 0.5$ J/m³; (a-2) $d_p = 2.5$ µm, $E_{ac} = 5$ J/m³; (a-3) $d_p = 2.5$ µm, $E_{ac} = 10$ J/m³; (b-1) $d_p = 10$ µm, $E_{ac} = 0.5$ J/m³; (b-2) $d_p = 10$ µm, $E_{ac} = 5$ J/m³; (b-3) $d_p = 10$ µm, $E_{ac} = 10$ J/m³.

wavelength) centered at the pressure node, located at $y = 4.25$ mm. The horizontal axis corresponds to the length of the acoustic agglomeration chamber ($L = 500$ mm).

As shown in Fig 11, as the acoustic energy density ($E_{ac}$) increases, particle migration toward the pressure node became more evident. At a low $E_{ac}$ value of 0.5 J/m³ (Figs 11(a-1) and (b-1)), the acoustic radiation force acting on the particles was weak, and all particles were carried through the chamber by the fluid flow. When $E_{ac}$ reached 5 J/m³ (Figs 11(a-2) and (b-2)), significant particle migration toward the pressure node was observed, leading to the formation of a concentrated particle band near the location ($x = 250$ mm). At $E_{ac} = 10$ J/m³ (Figs 11(a-2) and (b-2)), both particle sizes exhibited efficient agglomeration near the outlet. Larger particles ($d_p = 10$ µm), however, migrated more rapidly toward the pressure node, owing to stronger size-dependent acoustic radiation forces ($F_{ac} \propto d_p^3$).

**3.2.2. The particle spatial distribution in acoustic-flow field.** The spatial evolution of particle clouds ($d_p = 2.5$ µm and 10 µm) under varying acoustic energy densities ($E_{ac} = 0.5$, 5, and 10 J/m³) is presented in Fig 12. The simulation parameters were set to $f = 20$ kHz and $u_g = 0.75$ m/s. The vertical axis ($y$-direction) corresponds to acoustic wave propagation and contains two acoustic pressure nodes that act as particle accumulation zones. Three representative duct segments are illustrated: (inlet: $x = 0$–$20$ mm; mid-section: $x = 240$–$260$ mm; outlet: $x = 480$–$500$ mm).

Fig 12 shows that the particle ensemble, carried by the airflow along the $x$-direction, gradually migrates and accumulates at the acoustic pressure nodes due to acoustic radiation forces. As the acoustic energy density increases, the agglomeration effect became more pronounced. At $E_{ac} = 10$ J/m³, both $d_p = 2.5$ µm and 10 µm particles underwent effective agglomeration within the chamber (see Figs 12(a-3) and 12(b-3)). It was noteworthy that the agglomeration effect was stronger for larger particles, which experienced greater acoustic radiation forces.

**3.2.3. The profiles of dust particles concentration.** Fig 13 shows the distributions of particle number concentrations under different acoustic energy densities ($E_{ac}$). The vertical axis represents the percentage increment in particle concentration, defined as $(n-n_0)/n_0 \times 100\%$, where $n_0$ denotes the initial particle concentration. The horizontal axis indicates the $y$-coordinate across one acoustic wavelength ($\lambda$), with $y = 4.25$ mm and $12.75$ mm corresponding to the

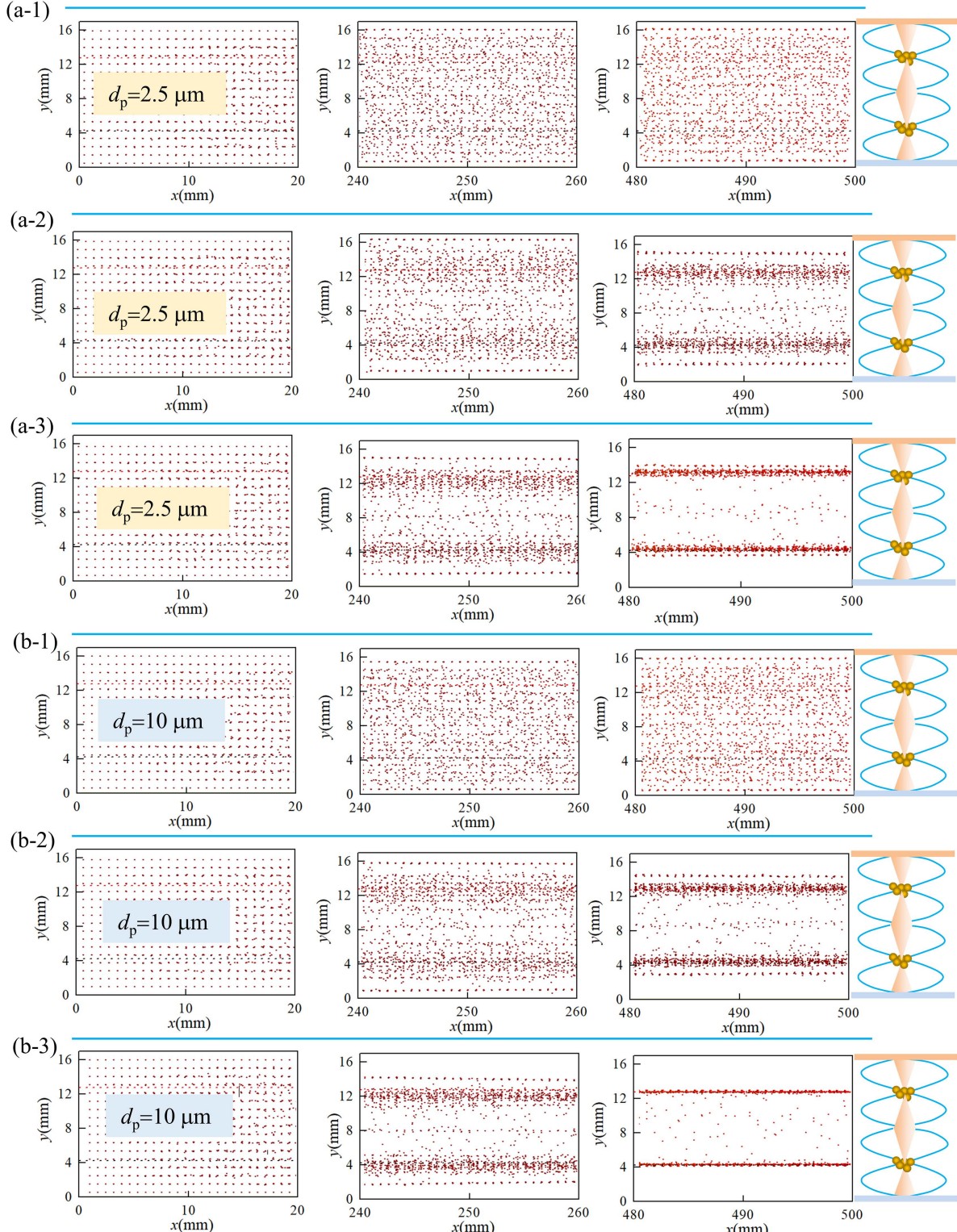

**Fig 12. Spatial distribution of dust particles at different ultrasonic energy densities.** (a-1) $d_p$ = 2.5 μm, $E_{ac}$ = 0.5 J/m³; (a-2) $d_p$ = 2.5μm, $E_{ac}$ = 5J/m³ (a-3) $d_p$ = 2.5μm, $E_{ac}$ = 10J/m³; (b-1) $d_p$ = 10μm, $E_{ac}$ = 0.5J/m³; (b-2) $d_p$ = 10μm, $E_{ac}$ = 5J/m³; (b-3) $d_p$ = 10 μm, $E_{ac}$ = 10J/m³.

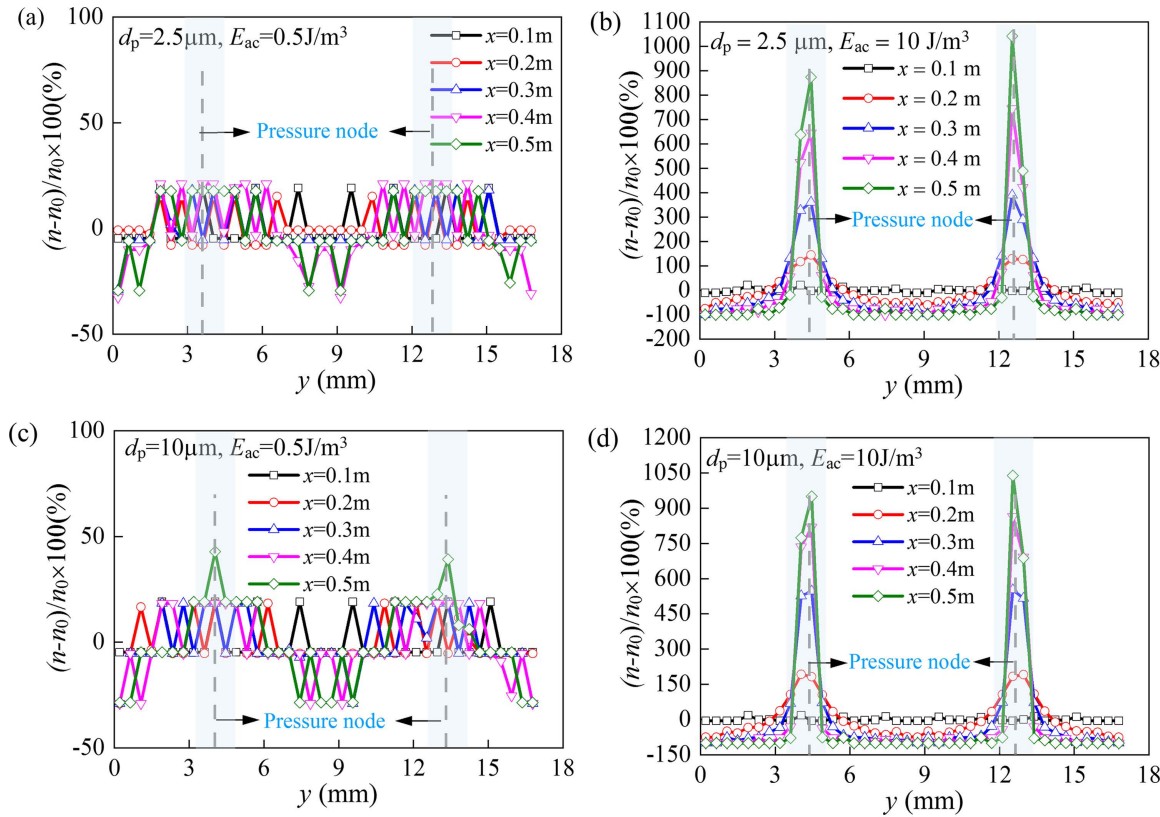

**Fig 13. Evolution of particle concentration distribution along the flow direction.** (a) $d_p = 2.5$ μm, $E_{ac} = 5$ J/m³; (b) $d_p = 2.5$ μm, $E_{ac} = 10$ J/m³; (c) $d_p = 10$ μm, $E_{ac} = 0.5$ J/m³; (d) $d_p = 10$ μm, $E_{ac} = 10$ J/m³.

pressure nodes of the standing wave. To elucidate the spatial evolution of particle concentration along the flow direction ($x$-axis), five cross-sections were selected for analysis: $x = 0.1$ m, 0.2 m, 0.3 m, 0.4 m, and 0.5 m.

Under weak acoustic energy density ($E_{ac} = 0.5$ J/m³, see Figs 13(a) and 13(b)), the particle concentration profiles remained nearly identical across all cross-sections, indicating minimal influence of the acoustic field on particle spatial distribution. Only a 25% enhancement in particle concentration was observed at pressure nodes ($y = 4.25$ mm and 12.75 mm), due to insufficient acoustic radiation forces. This result was consistent with the particle dynamics shown in the trajectory spectra (Fig 11) and the corresponding spatial distribution of particles (Fig 12). Progressive augmentation of $E_{ac}$ led to substantial concentration amplification at acoustic nodes. At $E_{ac} = 10$ J/m³, enrichment factors exceeded 900% (equivalented to a tenfold increase over initial concentrations) for both $d_p = 2.5$ μm and 10 μm particles at the downstream monitoring port ($x = 0.5$ m) (Figs 13(c) and 13(d)). According to particle agglomeration theory [45], the collision probability scales with the square of particle concentration. Thus, in this scenario, the acoustically induced particle collision probability was enhanced by two orders of magnitude.

**3.2.4. Droplet motion trajectories.** In this study, ultrafine droplets served as "agglomeration nuclei" in conjunction with superimposed standing-wave acoustic fields to enhance dust removal efficiency. Both ultrafine droplets and dust particles were introduced simultaneously into the agglomeration chamber, where they underwent acoustic agglomeration under identical standing-wave conditions. Accordingly, the subsequent analysis of droplet motion and agglomeration in high-frequency standing-wave fields employs the same parameters used for dust particles. Based on ultrasonic

atomization measurements, the generated droplets exhibited a size distribution ranging from 10 to 50 µm. Therefore, two representative droplet sizes ($d_w = 10$ µm and 50 µm) were selected for detailed analysis.

Fig 14 presents the trajectories of ultrafine droplets under varying acoustic energy densities ($E_{ac} = 0.5$–$10$ J/m³), with $f = 20$ kHz and $u_g = 0.75$ m/s. For $d_w = 10$µm droplets (Fig 14(a)), the movement patterns resembled those of similarly sized dust particles due to comparable acoustic radiation and drag forces. In contrast, larger droplets ($d_w = 50$ µm, Fig 14(b)) exhibited immediate migration toward the pressure nodes even at a low acoustic energy density $E_{ac} = 0.5$ J/m³ and developed an underdamped oscillating mode at $E_{ac} = 10$ J/m³ characterized by nodal traversals with decaying amplitudes — a phenomenon not observed in dust particle studies. A comparison between Fig 14 (b-3) and Fig 11 (b-3) reveals that $d_w = 50$ µm ultrafine droplets (serving as potential "agglomeration nuclei") formed clusters as early as at $x = 0.1$ m, whereas $d_p = 10$ µm dust particles aggregate around $x = 0.5$ m. It meant that dust and droplets failed to coalesce at the same position. Crucially, premature droplet arrival at pressure nodes promoted homogeneous droplet-droplet collisions, resulting to coalescence. This process formed larger droplets that precipitated prematurely, thereby impairing their function as agglomeration nuclei and significantly reducing the efficiency of dust-droplet hetero-agglomeration. These findings highlight the importance of matching droplet and dust particle sizes in practical applications to ensure optimal system performance.

**3.2.5. The profiles of droplets concentration.** Fig 15 presents the spatial evolution of droplet concentration profiles under the same acoustic conditions as in Fig 13. Only two practically significant cases involving small droplets with diameters of 10 µm and 20 µm were discussed.

It was evident that droplets exhibited stronger agglomeration than solid particles of the same size. At $E_{ac} = 0.5$ J/m³, 10 µm droplets showed a concentration enhancement of approximately 70% at the chamber outlet ($x = 0.5$ m, Fig 15(a)), which was four times higher than the 25% increase observed for dust particles under the same conditions. At a higher energy density ($E_{ac} = 10$ J/m³), the difference became more pronounced: 20 µm droplets exhibited a striking concentration

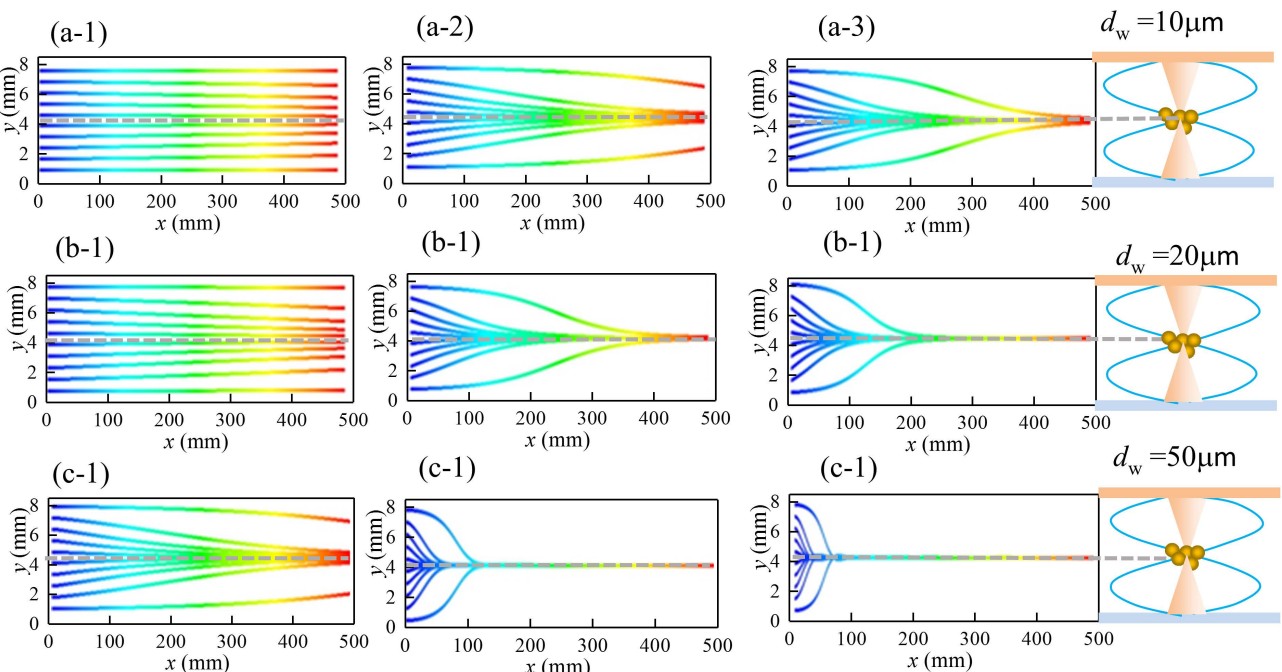

**Fig 14. Trajectories of droplets under different ultrasonic energy densities.** (a-1) $d_w = 10$ µm, $E_{ac} = 0.5$ J/m³; (a-2) $d_w = 10$ µm, $E_{ac} = 5$ J/m³; (a-3) $d_w = 10$ µm, $E_{ac} = 10$ J/m³; (b-1) $d_w = 20$ µm, $E_{ac} = 0.5$ J/m³; (b-2) $d_w = 20$ µm, $E_{ac} = 5$ J/m³; (b-3) $d_w = 20$ µm, $E_{ac} = 10$ J/m³; (c-1) $d_w = 50$ µm, $E_{ac} = 0.5$ J/m³; (c-2) $d_w = 50$ µm, $E_{ac} = 5$ J/m³; (c-3) $d_w = 50$ µm, $E_{ac} = 10$ J/m³.

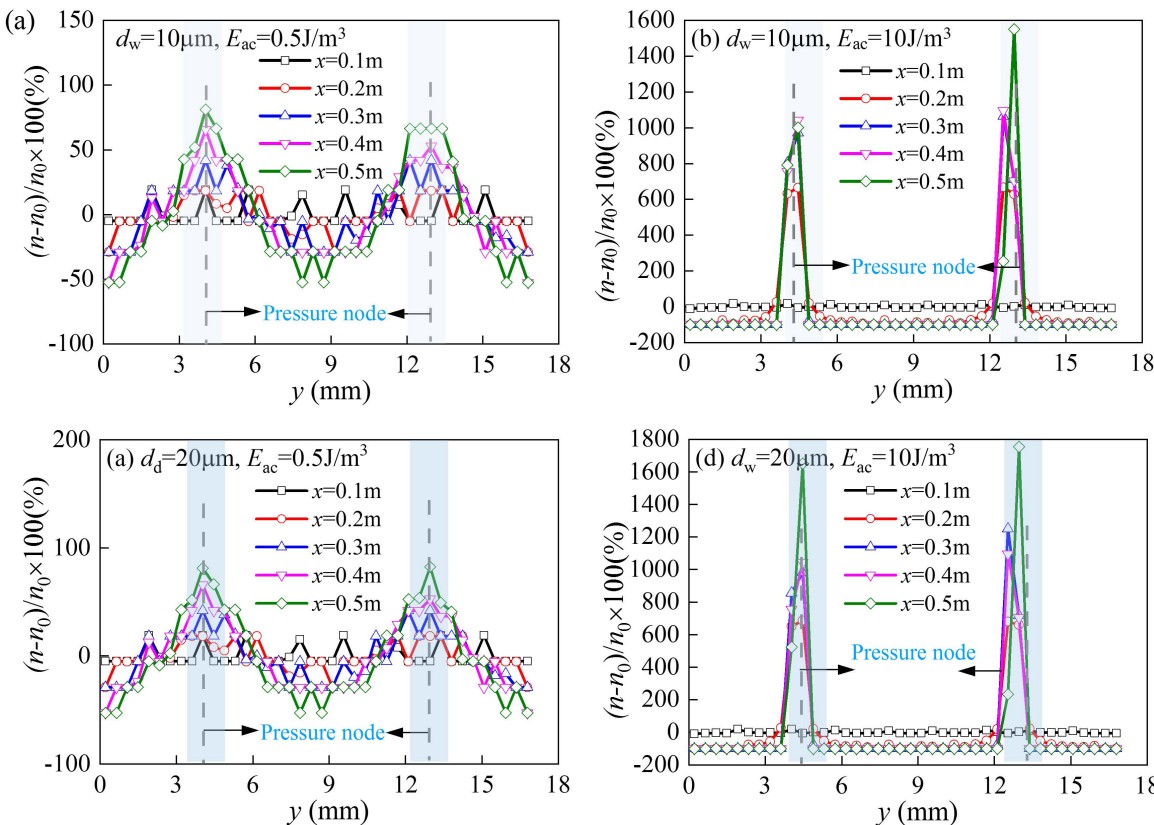

**Fig 15. Evolution of droplet concentration distributions.** (a) $d_w = 10$ μm, $E_{ac} = 5$ J/m³;(b) $d_w = 10$ μm, $E_{ac} = 2.5$ J/m³; (c) $d_w = 20$ μm, $E_{ac} = 0.5$ J/m³; (d) $d_w = 20$ μm, $E_{ac} = 10$ J/m³.

increase of 1600% ((Fig 15(d))). This enhanced agglomeration efficiency was attributed to the stronger acoustic radiation forces acting on liquid droplets, resulting from their distinct density and compressibility compared to solid particles. The results indicated that droplets in the 10–20 μm size range serve as optimal agglomeration nuclei, offering an ideal balance between migration kinetics and interaction probability while avoiding premature coagulation, which could otherwise reduce acoustic aggregation efficiency.

## 4. Conclusions and outlook

(1) This study demonstrates a clear synergistic effect in an integrated "ultrafine droplets + acoustic waves + high-pressure spray" system, which enhances respirable coal dust capture efficiency by more than 10% compared to conventional spray systems. In contrast, the configuration of "acoustic waves + high-pressure spray" alone shows only negligible improvement, underscoring the essential role of ultrafine droplets in achieving effective dust suppression.

(2) Two critical operational dependencies were identified to govern system performance. First, the acoustic power must exceed a critical threshold to initiate effective dust-droplet agglomeration under a given airflow velocity. Second, increased airflow velocities reduce particle residence time in the acoustic field, thereby weakening the synergistic effect. Consequently, optimal performance requires a careful balance between acoustic power and airflow velocity within a well-defined operational window.

(3) Numerical simulations provide mechanistic insight into the observed synergy, confirming that acoustic radiation forces drive both dust particles and droplets toward pressure nodes, forming localized high-concentration zones. These zones significantly promote collision and agglomeration. Under typical operating conditions (e.g., 0.75 m/s airflow, 10 J/m³ acoustic energy density), the local particle concentration at the nodes can increase by more than tenfold.

(4) A key finding is that effective agglomeration depends critically on the synchronized migration of dust and droplets to the acoustic nodes. Mismatched kinetics—wherein droplets reach the nodes faster than dust particles—can limit nucleation efficiency. Therefore, system performance can be optimized by aligning the migration rates of both droplets and dust particles through parameter tuning.

(5) It should be noted that this study has limitations, primarily stemming from the idealized conditions of the laboratory environment. In practical engineering contexts, such as coal mining roadways, wall surfaces consist of rough coal or rock strata rather than the smooth acrylic used in our experiments. Furthermore, background acoustic noise from operating machinery may interfere with the intended sound field, while phenomena like acoustic energy dissipation, wall reflection losses, and droplet evaporation—all unaccounted for in the current numerical model—can collectively influence the agglomeration process. Their omission may lead to an overestimation of capture efficiency in real-world scenarios. Future work should therefore incorporate these complexities to enhance the predictive accuracy and practical reliability of the simulation framework.

(6) Despite being contextualized in coal dust suppression, the fundamental mechanisms and synergistic principles revealed in this study—particularly acoustic-driven particle migration and enrichment for enhanced coagulation—hold significant potential for broader particulate control applications. For instance, in tunnel fire rescue operations, the combined use of acoustic waves and spraying could achieve rapid dust settlement in smoke-filled sections. In powder processing facilities, such as those in the pharmaceutical or chemical industries, this approach could effectively control highly diffusive fine product dust, thereby improving workplace safety and reducing product loss. Additionally, the system presents a promising solution for managing challenging fine particulates like fly ash from power plants or metallurgical fumes, which often pose difficulties for conventional removal systems. The key insights gained regarding acoustic power thresholds, airflow matching, and particle-droplet kinetics provide a universal framework for optimizing such acoustic-agglomeration systems across varied scales and dust types.

## Supporting information

**S1 File. Raw data.**
(XLSX)

## Acknowledgments

The authors would like to thank Prof. Haiqiao Wang for providing valuable comments on the manuscript.

## Author contributions

**Conceptualization:** Shiqiang Chen, Heqing Liu.

**Data curation:** Hui Yang, Chunyu Liu.

**Formal analysis:** Shixian Wu.

**Funding acquisition:** Hui Zhu.

**Investigation:** Yongping Chen.

**Methodology:** Shixian Wu, Hui Zhu, Hui Yang, Shiqiang Chen.

**Project administration:** Hui Zhu.

**Resources:** Can Qi, Shiqiang Chen.

**Software:** Can Qi.

**Supervision:** Heqing Liu.

**Validation:** Chunyu Liu.

**Visualization:** Can Qi, Hui Yang, Chunyu Liu.

**Writing – original draft:** Shixian Wu.

**Writing – review & editing:** Hui Zhu.

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
