## [Decision Letter · Decision Letter 0]

29 Sep 2025

Dear Dr. Zhu,

We look forward to receiving your revised manuscript.

Kind regards,

Antonio Riveiro Rodríguez, PhD

Academic Editor

PLOS ONE

Journal Requirements:

3. Please note that PLOS One has specific guidelines on code sharing for submissions in which author-generated code underpins the findings in the manuscript. In these cases, we expect all author-generated code to be made available without restrictions upon publication of the work. Please review our guidelines at https://journals.plos.org/plosone/s/materials-and-software-sharing#loc-sharing-code and ensure that your code is shared in a way that follows best practice and facilitates reproducibility and reuse.

5. Please remove your figures from within your manuscript file, leaving only the individual TIFF/EPS image files, uploaded separately. These will be automatically included in the reviewers’ PDF**.**

6. Please upload a copy of Supporting Information Figures 1, 2, 3, 4, 5, 6, 7, 8, 9, 10, 11, and 12, which you refer to in your text on page 22, 23.

7. Please upload a copy of Supporting Information Tables 1, 2, and 3, which you refer to in your text on page 22, 23.

8. We notice that your supplementary figures are uploaded with the file type 'Figure'. Please amend the file type to 'Supporting Information'. Please ensure that each Supporting Information file has a legend listed in the manuscript after the references list.

Reviewers' comments:

Reviewer's Responses to Questions

**Comments to the Author**

1. Is the manuscript technically sound, and do the data support the conclusions?

Reviewer #1: Yes

Reviewer #2: Yes

2. Has the statistical analysis been performed appropriately and rigorously?

Reviewer #1: Yes

Reviewer #2: Yes

3. Have the authors made all data underlying the findings in their manuscript fully available?

Reviewer #1: Yes

Reviewer #2: Yes

4. Is the manuscript presented in an intelligible fashion and written in standard English?

Reviewer #1: Yes

Reviewer #2: Yes

Reviewer #1: Review

Article: Enhanced dust removal via the synergy of a standing wave acoustic field and high-pressure spray: an integrated experimental and numerical study

This article is devoted to a comprehensive study of a new method proposed by the authors for removing coal dust in mines. This method involves the preliminary introduction of coagulation nuclei (highly dispersed water droplets) and accelerated coal dust coagulation under conditions of a standing acoustic wave and a high-pressure spray jet. The proposed method is interesting, and the authors' conscientious approach to the research methodology is also noteworthy. However, the text of the manuscript raises many questions (which I hope the authors will address).

My questions and comments will be presented in the order they appear in the manuscript:

1. Abstract (lines 30, 31): "nozzle orifice diameter" – It's unclear here that this refers to the high-pressure nozzle. "acoustic power density" – It's also unclear here that this refers to the acoustic power when creating a standing wave (since you also have ultrasonic atomization). Just check the text.

2. Introduction. Line 68. Please note the article that presents new methods of ultrasonic atomization (Shalunov, A., Kudryashova, O., Khmelev, V., Genne, D., Terentiev, S., & Nesterov, V. (2024). Innovative ultrasonic spray methods for indoor disinfection. Applied System Innovation, 7(6), 126.)

3. Line 82. You may not have seen the new article (Xu, R. C., Sharma, A. K., Ozdemir, E., Miwa, S., & Suzuki, S. (2024). Experimental investigation on effective aerosol scavenging using different spray configurations with pre-injection of water mist for Fukushima Daiichi decommissioning. Nuclear Science and Techniques, 35(5), 42)

4. Line 83. There is an article about the mechanisms of agglomeration and sedimentation of particles in acoustic fields (Kudryashova, O., Antonnikova, A., Korovina, N., & Akhmadeev, I. (2015). Mechanisms of aerosol sedimentation by acoustic field. Archives of Acoustics, 485-489)

5. Experimental setup. Line 114. It's unclear here what kind of droplets—those generated by the ultrasonic nebulizer or the nozzle? The Malvern isn't visible in the experimental setup diagram. It would be interesting to see the droplet sizes over time. Also, you didn't mention the ultrasonic nebulizer. What kind is it? What size droplets does it generate? What size droplets do the nozzles generate (perhaps this is also important)?

6. Coal dust preparation. Line 124. It's unclear what high atomization efficiency means and why the water consumption is low. Compared to what? Moreover, the same water consumption will be given for all orifice sizes (180 ml/min).

7. Fig. 3. Recommendation: provide statistical characteristics, average diameters. Also: samples were taken at the inlet and outlet, #1 and #2. I wonder if there was a difference in dust size in these samples?

8. Model description. Line 152. How scalable are these dimensions? Why were these chosen (significantly smaller than in the experiment or in real tunnel conditions).

9. Dynamics of dust/droplet motion. Line 167. To what extent can Brownian motion be neglected? Under what conditions is this possible?

10. 3.1.1 Effects of nozzle orifice diameter. Line 229. The flow rate was the same for three nozzle hole diameters. What changed depending on these hole diameters? (I'm assuming it was the droplet size, or maybe not?) It's unclear how the hole sizes themselves could influence the processes under consideration. It would be interesting to discuss this further.

11. Discussion of Figure 5. Figure 5c is certainly impressive. But looking at Figures 5a and 5b, it's clear that the efficiency increase is occurring from an already good baseline—80% or more. If the system without acoustic coagulation already works well, how justified is a standing wave generator, especially in a mine's natural environment, if it increases dust removal efficiency from 80% to 90%?

12. 3.1.3 Effects of airflow velocity It would be useful to plot a graph of overall efficiency (not by particle size, but for the entire population, according to their size distribution) as a function of flow rate. This would show at what flow rates the efficiency of acoustic coagulation continues to increase, and at what rates it no longer does.

13. Line 274. I'm afraid increasing the acoustic power won't help. Indeed, the time spent in the acoustic zone is more important here, so the lower the ventilation speed, the greater the effect. Perhaps increasing the rate of ultrafine droplet generation while simultaneously increasing the airflow speed might help?

14. Captions for Figure 7. Here you need to indicate what air flow speed each figure corresponds to.

15. 3.2.5. The profiles of droplets concentration. Line 365. You only considered two droplet sizes and two levels of impact intensity. It would be interesting to see, for example, a graph of droplet concentration at a pressure node for a single impact intensity as a function of droplet size. Perhaps your conclusion that 10-20 µm are the optimal agglomeration nuclei is incorrect? Perhaps, for example, nuclei of 5 or 30 µm would be better?

16. Line 390. Could you be more specific here, with numbers – which drops are large? Which are optimal?

17. At the end of the Conclusion, it would be useful to discuss the potential practical applications of the results of this work. How feasible is it to create similar installations in mines? How economically feasible is this (especially considering the generally modest increase in dust removal efficiency)? On the other hand, your work could have broader application, not only (and not primarily) for coal dust removal, but also in other areas where hazardous fine dust is present.Also, please clearly state the fundamental scientific novelty of your results.

Overall, the article is interesting, and I eagerly await its publication. However, the text needs some work to clarify any unclear points and inaccuracies. I wish the authors success with the publication.

Reviewer #2: 1. Applicability and scalability of experimental conditions

The laboratory-scale tunnel model provides valuable insights into the synergistic mechanisms of acoustic waves and sprays. However, underground mining environments present far more complex conditions, including larger airflow scales, turbulence, and irregular geometries. The current work would benefit from a more explicit discussion of how the experimental results can be reasonably extrapolated to real-world mining operations, as well as the potential limitations of such extrapolation. Highlighting the constraints of laboratory findings and outlining a pathway for future large-scale validation would considerably strengthen the practical impact of this study.

2. Justification of parameter selection

The choice of critical experimental parameters—such as ultrasonic frequency (20 kHz), nozzle orifice size (0.4–0.8 mm), and acoustic power (60–180 W)—is briefly described but lacks sufficient justification. Readers may find it difficult to assess whether these parameters are representative of actual mine dust suppression systems or were chosen arbitrarily. It is recommended that the authors provide additional references or results from preliminary trials to clarify why these ranges are appropriate. Such information would help ensure the generalizability and credibility of the reported findings.

3. Simplification in numerical modeling

The numerical model reasonably captures the essential physics of particle–droplet interactions in a standing-wave acoustic field. However, several key factors—such as acoustic energy dissipation, wall reflection losses, and droplet evaporation—were neglected. These simplifications may lead to an overestimation of agglomeration efficiency. It would strengthen the paper if the authors acknowledged these assumptions in greater detail and discussed their potential effects on the simulation outcomes. In addition, outlining future directions to incorporate these complexities would enhance the robustness and reliability of the modeling approach.

4. Lack of statistical and uncertainty analysis

The improvements in dust removal efficiency are mainly reported as average values without sufficient treatment of variability or statistical reliability. For instance, Figures 5–7 present performance enhancements but do not include error bars, confidence intervals, or significance testing. This omission reduces the persuasiveness of the results, especially given the potential variability in dust and droplet interactions. It is recommended that the authors conduct uncertainty analysis and add statistical indicators to their figures. Such additions would provide readers with a clearer understanding of the robustness of the experimental findings.

5. Structure and presentation improvements

While the manuscript is generally well-written, some sections—particularly the Results and Discussion—contain redundant or loosely connected descriptions. For example, certain figure captions (e.g., Figures 5–7) repeat information already provided in the text, which can make the narrative less concise. I suggest tightening the link between the figures and the discussion, emphasizing comparative insights rather than restating data. Furthermore, the Conclusions could be improved by presenting a clearer summary of the engineering implications and practical application potential. This would make the paper more compelling to both academic and industrial audiences.

**Do you want your identity to be public for this peer review?** For information about this choice, including consent withdrawal, please see our Privacy Policy

Reviewer #1: No

Reviewer #2: No

---

## [Author Response · Author response to Decision Letter 1]

19 Nov 2025

Detailed Response to the Editor

Comment 6: Please upload a copy of Supporting Information Figures 1, 2, 3, 4, 5, 6, 7, 8, 9, 10, 11, and 12, which you refer to in your text on page 22, 23.

Response: We have now uploaded the Supporting Information Figures 1–15 as a supporting information file.

Comment 7: Please upload a copy of Supporting Information Tables 1, 2, and 3, which you refer to in your text on page 22, 23.

Response: We have now uploaded the Supporting Information Tables 1, 2, and 3 as a supporting information file.

Comment 8: We notice that your supplementary figures are uploaded with the file type 'Figure'. Please amend the file type to 'Supporting Information'. Please ensure that each Supporting Information file has a legend listed in the manuscript after the references list.

Response: Thank you for highlighting this issue. We have corrected the file types to "Supporting Information" as instructed. Furthermore, we have ensured that a descriptive legend for each supporting information file is now included in the manuscript immediately after the reference list.

Comment 9: If the reviewer comments include a recommendation to cite specific previously published works, please review and evaluate these publications to determine whether they are relevant and should be cited. There is no requirement to cite these works unless the editor has indicated otherwise.

Response: We have thoroughly evaluated the relevant publications suggested by the reviewers and have incorporated appropriate citations into the manuscript where deemed necessary.

All other comments and requests for confirmation from the Editorial Office have been addressed meticulously, and corresponding revisions have been made throughout the manuscript.

Detailed Response to the reviewers’ comments:

Reviewer #1:

General Comments:

This article is devoted to a comprehensive study of a new method proposed by the authors for removing coal dust in mines. This method involves the preliminary introduction of coagulation nuclei (highly dispersed water droplets) and accelerated coal dust coagulation under conditions of a standing acoustic wave and a high-pressure spray jet. The proposed method is interesting, and the authors' conscientious approach to the research methodology is also noteworthy. However, the text of the manuscript raises many questions (which I hope the authors will address).

My questions and comments will be presented in the order they appear in the manuscript:

Specific Comments:

Comment 1. Abstract (lines 30, 31): "nozzle orifice diameter" – It's unclear here that this refers to the high-pressure nozzle. "acoustic power density" – It's also unclear here that this refers to the acoustic power when creating a standing wave (since you also have ultrasonic atomization). Just check the text.

Response� We thank the reviewer for their careful reading. The term "nozzle orifice diameter" in the abstract (lines 30–31) refers to the diameter of the high-pressure nozzle and that the "acoustic power" represents the power utilized to generate the standing wave field. We have clarified this in the revised manuscript.

Comment 2. Introduction. Line 68. Please note the article that presents new methods of ultrasonic atomization (Shalunov, A., Kudryashova, O., Khmelev, V., Genne, D., Terentiev, S., & Nesterov, V. (2024). Innovative ultrasonic spray methods for indoor disinfection. Applied System Innovation, 7(6), 126.)

Response� We appreciate the reviewer for recommending the relevant literature. The cited article primarily introduces novel design methods for ultrasonic atomization devices. While it does not directly address the acoustic dust suppression mechanisms central to our study, the design approaches presented are valuable for our future work aimed at practical engineering applications. Therefore, we consider the direct relevance to this paper limited and have not cited it in this context, but we will certainly draw on its insights in subsequent research.

Comment 3. Line 82. You may not have seen the new article (Xu, R. C., Sharma, A. K., Ozdemir, E., Miwa, S., & Suzuki, S. (2024). Experimental investigation on effective aerosol scavenging using different spray configurations with pre-injection of water mist for Fukushima Daiichi decommissioning. Nuclear Science and Techniques, 35(5), 42)

Response� We thank the reviewer for bringing this recent publication to our attention. We had overlooked this article and have now included it as a reference in the revised manuscript (see reference 22).

Comment 4. Line 83. There is an article about the mechanisms of agglomeration and sedimentation of particles in acoustic fields (Kudryashova, O., Antonnikova, A., Korovina, N., & Akhmadeev, I. (2015). Mechanisms of aerosol sedimentation by acoustic field. Archives of Acoustics, 485-489)

Response� We are grateful to the reviewer for suggesting this relevant study. The recommended paper, based on a mathematical model of aerosol settling in an acoustic field, clearly identifies two key mechanisms: enhanced gravitational settling due to particle agglomeration and acoustic radiation pressure effects. The findings provide useful insights for understanding the synergistic dust suppression mechanism of spray and acoustic waves, and we have cited this work (see reference 35) in the revised manuscript.

Comment 5. Experimental setup. Line 114. It's unclear here what kind of droplets—those generated by the ultrasonic nebulizer or the nozzle? The Malvern isn't visible in the experimental setup diagram. It would be interesting to see the droplet sizes over time. Also, you didn't mention the ultrasonic nebulizer. What kind is it? What size droplets does it generate? What size droplets do the nozzles generate (perhaps this is also important)?

Response� We thank the reviewer’s comments. We have clearly specified the types of droplets used in the revised manuscript. During the experiments, we attempted to observe the temporal and spatial variations in droplet size to clarify the agglomeration behavior between droplets and dust under acoustic influence. However, due to obstruction of the optical path by water mist deposition on the pipe wall (despite the use of highly transparent acrylic material), the Malvern instrument could not accurately track changes in droplet size.

To address this limitation, we separately measured the droplet size distributions generated by the ultrasonic atomizer and the high-pressure spray using the Malvern instrument prior to the formal experiments. These measurement results have been added to the revised manuscript (see 2.14 Droplet Atomization Characteristics), and the Malvern instrument is shown in Fig 4.

Comment 6. Coal dust preparation. Line 124. It's unclear what high atomization efficiency means and why the water consumption is low. Compared to what? Moreover, the same water consumption will be given for all orifice sizes (180 ml/min).

Response� The experimental setup in this study is a scaled-down model of a real tunnel, requiring only a small spray coverage area. Hence, a spray device with low water consumption (maximum 180 mL/min) was selected.

The terms "low water consumption" and "high atomization efficiency" refer specifically to a comparison with the high-flow high-pressure nozzles (approx. 1 L/min) commonly used in large-scale dust suppression in underground coal mines. Higher atomization efficiency helps generate finer droplets, which is crucial for the acoustic-spray synergistic dust removal process.

Given the importance of droplet characteristics, we have supplemented the revised manuscript with detailed spray droplet parameters (see Fig 5).

Comment 7. Fig. 3. Recommendation: provide statistical characteristics, average diameters. Also: samples were taken at the inlet and outlet, #1 and #2. I wonder if there was a difference in dust size in these samples?

Response� We thank the reviewer for the suggestion. The average diameter of the coal dust particles has been added to Figure 3 in the revised manuscript. In this study, the overall dust removal efficiency was evaluated by measuring the weight of dust collected at the inlet and outlet. Since acoustic waves and spray droplets influence dust particles of different sizes differently—for instance, spray capture is more effective for larger particles—the particle size distribution of samples collected at the inlet (#1) and outlet (#2) may indeed differ. The primary objective of this work was to validate the feasibility of the acoustic-spray synergy and clarify the underlying mechanism; therefore, a detailed analysis of the changes in particle size between sampling points was not conducted. We agree that such analysis would be valuable for further mechanistic understanding and have included it in our planned follow-up work.

Comment 8. Model description. Line 152. How scalable are these dimensions? Why were these chosen (significantly smaller than in the experiment or in real tunnel conditions).

Response� We thank the reviewer’s comment. This study was designed as a preliminary investigation into the feasibility and mechanical principles of the acoustic-spray synergistic dust reduction approach. A small-scale experimental model was adopted to allow well-controlled experimental conditions and manageable costs. Although the model dimensions are smaller than those of a real tunnel, the fundamental physical mechanisms revealed—such as acoustic agglomeration and particle-droplet interactions—are consistent with real conditions. We fully agree that the ultimate translation and application of this technology will require large-scale tests and numerical simulations under real tunnel conditions, which entail considerably higher costs (e.g., developing high-power acoustic devices and performing long-term measurements). This represents an important direction for our future research.

Comment 9. Dynamics of dust/droplet motion. Line 167. To what extent can Brownian motion be neglected? Under what conditions is this possible?

Response� We appreciate the reviewer’s observation. Given that the flow in the tunnel model is turbulent, and the intensity of turbulent fluctuations is much greater than that of particle Brownian motion, the effect of Brownian motion on particle behavior was considered negligible. A brief justification of this assumption has been added to the revised manuscript�see page 9, lines 196-198

Comment 10. 3.1.1 Effects of nozzle orifice diameter. Line 229. The flow rate was the same for three nozzle hole diameters. What changed depending on these hole diameters? (I'm assuming it was the droplet size, or maybe not?) It's unclear how the hole sizes themselves could influence the processes under consideration. It would be interesting to discuss this further.

Response� As correctly noted by the reviewer, the nozzle orifice diameter influences the resulting droplet size. Under the same flow rate conditions, a smaller orifice generally yields finer droplets. Data on droplet size distributions and spray angles for different nozzle orifice diameters have been included in the revised manuscript (see 2.14 Droplet Atomization Characteristics).

Comment 11. Discussion of Figure 5. Figure 5c is certainly impressive. But looking at Figures 5a and 5b, it's clear that the efficiency increase is occurring from an already good baseline—80% or more. If the system without acoustic coagulation already works well, how justified is a standing wave generator, especially in a mine's natural environment, if it increases dust removal efficiency from 80% to 90%?

Response� Based on the current experimental results, the incorporation of acoustic agglomeration improved the dust removal efficiency from 80% to 90%. Although the absolute gain is 10%, this improvement is nonetheless encouraging. We believe that further systematic parameter optimization could enhance the synergistic dust removal performance. Our subsequent research will include trials under real mine conditions to validate the technical feasibility of this approach.

Comment 12. 3.1.3 Effects of airflow velocity It would be useful to plot a graph of overall efficiency (not by particle size, but for the entire population, according to their size distribution) as a function of flow rate. This would show at what flow rates the efficiency of acoustic coagulation continues to increase, and at what rates it no longer does.

Response� We thank the reviewer for the suggestion. Data on the acoustic-spray synergistic dust removal efficiency under different flow velocities have been added to the revised manuscript (see Fig 9. Dust removal efficiency as a function of air velocity).

Comment 13. Line 274. I'm afraid increasing the acoustic power won't help. Indeed, the time spent in the acoustic zone is more important here, so the lower the ventilation speed, the greater the effect. Perhaps increasing the rate of ultrafine droplet generation while simultaneously increasing the airflow speed might help?

Response� Our findings indicate that a longer residence time of particles in the acoustic field leads to better agglomeration with ultrafine droplets. Thus, a lower air velocity favors the agglomeration effect. We have modified the relevant description accordingly.

Comment14. Captions for Figure 7. Here you need to indicate what air flow speed each figure corresponds to.

Response� Thanks for the reviewer’s careful review. The corresponding airflow velocities have now been indicated for each subplot in Figure 7.

Comment 15. 3.2.5. The profiles of droplets concentration. Line 365. You only considered two droplet sizes and two levels of impact intensity. It would be interesting to see, for example, a graph of droplet concentration at a pressure node for a single impact intensity as a function of droplet size. Perhaps your conclusion that 10-20 µm are the optimal agglomeration nuclei is incorrect? Perhaps, for example, nuclei of 5 or 30 µm would be better?

Response� We appreciate the reviewer’s insightful comment. Based on simulated motion trajectories of particles and ultrafine droplets in the acoustic field, we hypothesize that when the size of the agglomeration nuclei (i.e., the ultrafine droplets) is similar to that of the dust particles, they can more synchronously migrate to the pressure nodes of the standing wave, thereby achieving efficient agglomeration. Since the dust particles used in our experiments were concentrated around 10 μm, we speculate that droplets in the 10–20 μm range may represent an optimal size for the nuclei. Excessively small droplets are prone to evaporation, reducing the number of effective nuclei, while much larger droplets may reach the nodes earlier and undergo self-agglomeration. To simplify the model, droplet evaporation was not considered in this study. In future work, we aim to improve the model to provide more accurate guidance on the design of agglomeration nuclei size.

Comment16. Line 390. Could you be more specific here, with numbers – which drops are large? Which are optimal?

Response� The term "droplets" here specifically refers to the ultrafine mist that serves as coagulation nuclei in the acoustic agglomeration process, and their size is described relative to the coal dust particles. The Conclusion section has been revised to highlight the scientific novelty and engineering implications of our work.

Comment 17. At the end of the Conclusion, it would be useful to discuss the potential practical applications of the results of this work. How feasible is it to create similar installations in mines? How economically feasible is this (especially considering the generally modest increase in dust removal efficiency)? On the other hand, your work could have broader application, not only (and not primarily) for coal dust removal, but also in other areas where hazardous fine dust is present. Also, please clearly state the fundamental scientific novelty of your results.

Response� We sincerely thank the reviewer for their valuable suggestions and positive recognition of our work. Building on the understanding gained in this study regarding the acoustic-spray synergy, we are now considering implementing this technique in a real mining environment to further evaluate its practical feasibility and economic v

---

## [Decision Letter · Decision Letter 1]

1 Dec 2025

Enhanced dust removal via the synergy of a standing wave acoustic field and high-pressure spray: an integrated experimental and numerical study

PONE-D-25-48254R1

Dear Dr. Zhu,

We’re pleased to inform you that your manuscript has been judged scientifically suitable for publication and will be formally accepted for publication once it meets all outstanding technical requirements.

Kind regards,

Antonio Riveiro Rodríguez, PhD

Academic Editor

PLOS ONE

Reviewers' comments:

Reviewer's Responses to Questions

**Comments to the Author**

Reviewer #1: All comments have been addressed

Reviewer #2: All comments have been addressed

2. Is the manuscript technically sound, and do the data support the conclusions?

Reviewer #1: Yes

Reviewer #2: Yes

3. Has the statistical analysis been performed appropriately and rigorously?

Reviewer #1: Yes

Reviewer #2: Yes

4. Have the authors made all data underlying the findings in their manuscript fully available?

Reviewer #1: Yes

Reviewer #2: Yes

5. Is the manuscript presented in an intelligible fashion and written in standard English?

Reviewer #1: Yes

Reviewer #2: Yes

Reviewer #1: The authors have thoroughly addressed all the points raised in the review, providing detailed and satisfactory responses to each comment. They have made substantial revisions to the manuscript, including clarifying key terms, adding necessary experimental data and justifications, incorporating relevant citations, improving figure presentations with statistical indicators, and explicitly discussing the limitations, scientific novelty, and potential applications of their work. The revised manuscript is now significantly strengthened and well-supported, and I am pleased to recommend it for publication.

Reviewer #2: (No Response)

**Do you want your identity to be public for this peer review?** For information about this choice, including consent withdrawal, please see our Privacy Policy

Reviewer #1: **Yes: ** Olga Kudryashova

Reviewer #2: No

---

## [Editor Report · Acceptance letter]

PONE-D-25-48254R1

PLOS One

Dear Dr. Zhu,

I'm pleased to inform you that your manuscript has been deemed suitable for publication in PLOS One. Congratulations! Your manuscript is now being handed over to our production team.

Kind regards,

on behalf of

Dr. Antonio Riveiro Rodríguez

Academic Editor

PLOS One